# Somno-Art Software identifies pathology-induced changes in sleep parameters similarly to polysomnography

**Laurie Thiesse**[1], **Luc Staner**[2], **Patrice Bourgin**[3,4], **Henri Comtet**[3,4], **Gil Fuchs**[1], **Debora Kirscher**[1], **Thomas Roth**[5], **Jean Yves Schaffhauser**[1], **Jay B. Saoud**[6,7], **Antoine U. Viola**[1] *

1 PPRS, Colmar, France, 2 Unité d'exploration des Rythmes Veille Sommeil, Centre Hospitalier de Rouffach, Rouffach, France, 3 Sleep Disorders Center & CIRCSom (International Research Center for ChronoSomnology), Strasbourg University Hospital, Strasbourg, France, 4 Institute for Cellular and Integrative Neurosciences, CNRS UPR 3212, Strasbourg, France, 5 Sleep Disorders Center, Henry Ford Hospital, Detroit, MI, United States of America, 6 PPRS Research Inc., Groton, Massachusetts, United States of America, 7 PPDA, LLC, Boston, Massachusetts, United States of America

* avi@pprs-research.com

**Data Availability Statement:** The data underlying the results presented in the study is available as a Supporting information file (PDF).

## Abstract

Polysomnographic sleep architecture parameters are commonly used to diagnose or evaluate treatment of sleep disorders. Polysomnography (PSG) having practical constraints, the development of wearable devices and algorithms to monitor and stage sleep is rising. Beside pure validation studies, it is necessary for a clinician to ensure that the conclusions drawn with a new generation wearable sleep scoring device are consistent to the ones of gold standard PSG, leading to similar interpretation and diagnosis. This paper reports on the performance of Somno-Art Software for the detection of differences in sleep parameters between patients suffering from obstructive sleep apnea (OSA), insomniac or major depressive disorder (MDD) compared to healthy subjects. On 244 subjects (n = 26 healthy, n = 28 OSA, n = 66 insomniacs, n = 124 MDD), sleep staging was obtained from PSG and Somno-Art analysis on synchronized electrocardiogram and actimetry signals. Mixed model analysis of variance was performed for each sleep parameter. Possible differences in sleep parameters were further assessed with Mann-Whitney U-test between the healthy subjects and each pathology group. All sleep parameters, except N1+N2, showed significant differences between the healthy and the pathology group. No significant differences were observed between Somno-Art Software and PSG, except a 3.6±2.2 min overestimation of REM sleep. No significant interaction 'group'*'technology' was observed, suggesting that the differences in pathologies are independent of the technology used. Overall, comparable differences between healthy subjects and pathology groups were observed when using Somno-Art Software or polysomnography. Somno-Art proposes an interesting valid tool as an aid for diagnosis and treatment follow-up in ambulatory settings.

**Funding:** This study was supported by PPRS SAS. There was no additional external funding received for this study. The funder provided support in the form of salaries for authors LT, DK, GF, AUV. JYS and JBS participated in the study design, data collection and analysis, decision to publish, and preparation of the manuscript. The specific roles of these authors are articulated in the 'author contributions' section.

**Competing interests:** I have read the journal's policy and the authors of this manuscript have the following competing interests: This study was supported by PPRS SAS. JYS and JBS are shareholders of PPRS SAS. LT, GF, DK and AUV, are employees of PPRS SAS. JBS is a payed consultant of PPRS Research Inc. JYS is shareholder of V-WATCH SA. PB, HC and LS are working in hospitals that have received monetary contribution for running the healthy study 2 and the OSA study. The remaining authors declare that the research was conducted in the absence of any commercial or financial relationships that could be construed as a potential conflict of interest. This does not alter our adherence to PLOS ONE policies on sharing data and materials.

# Introduction

Sleep continuity and architecture parameters are commonly used to diagnose sleep disorders or to evaluate pharmacological and behavioral interventions to treat sleep disorders. Insomnia for example is characterized by prolonged sleep latency (SL), increased wake after sleep onset and lower sleep efficiency [1,2], while patients suffering from depression often present increased REM sleep time and shortened REM sleep latency [3–6]. The gold standard to characterize sleep continuity and architecture disturbances is, to date, polysomnography (PSG). However this time-consuming method is cumbersome, patient burden and costly with limited access or total lack of availability in some areas. To enhance the access to sleep explorations, some new wearable and user-friendly technologies, are intended to characterize sleep stages based on cardiac activity and body movement in ambulatory settings. Indeed, sleep stages are dependent on the balance of the autonomic nervous activity, parasympathetic tone being predominant during slow wave sleep while sympathetic tone being predominant during REM sleep [7,8]. Somno-Art Software, a cardiac-based sleep scoring algorithm, has recently been evaluated in healthy subjects and pathological sleep profiles [9,10]. Good and excellent intra-class correlation (according to Cicchetti [11] cutoffs) between Somno-Art Software and PSG for standard sleep architecture and continuity parameters were found in healthy subjects and in patients suffering from insomnia, obstructive sleep apnea (OSA) or major depressive disorder (MDD). Moreover, the scoring provided by Somno-Art Software showed a robust inter-scorer reliability in the range of visual scorers [12].

Thus, treating clinicians may also be interested in the performance of Somno-Art Software in detecting differences in sleep parameters characteristic in patients, leading to similar interpretation and diagnosis when using the sleep staging from the Somno-Art Software or gold standard PSG.

To do so, the differences of standard sleep parameters extracted from the sleep recordings coming from healthy subjects and patients suffering from OSA, insomnia or MDD were compared. It was hypothesized that comparable findings would be observed with the visual scoring of PSG recordings and the Somno-Art Software analysis.

# Methods

## Dataset

**Source studies.** The present dataset includes 6 studies and comprises data coming from both healthy subjects (two studies) and sleep disordered patients (one study with OSA patients, one with chronic insomniacs and two with depressed patients). The primary objective of the first healthy subjects study was to investigate relationship between daytime activity and night sleep structure and the impact of noise on sleep pattern. For the second one, the primary objective was to investigate the effect of light on sleep, wake, electroencephalogram (EEG) and cognitive performances as a function of homeostatic sleep drive. All recorded nights from these two healthy subjects' studies were included in the dataset. The OSA study included patients diagnosed with OSA syndrome during a clinical evaluation night. The insomniac study and the 2 depression studies primary objectives were to evaluate the efficacy, safety and tolerability of investigational drugs. Only pre-treatment nights were included in the dataset. For all studies included in the present analysis and before undergoing sleep recordings, a standard screening of patients and healthy subjects' health status was done. More information on the protocol descriptions are detailed in the S1 Appendix.

All study protocols were approved by institutional review boards (IRB) (Healthy study 1: Arztekammer Berlin, EudraCT N°: 2012-001043-44; Healthy study 2: Comité de protection

des personnes est IV, RCB N˚: 2014-A00795-42; OSA study: Comité de protection des personnes est IV, RCB N˚: 2014-A00382-45; Insomniac study: local IRB (USA (Florida), Germany (Berlin, Hamburg, Schwerin), Netherlands (Leiden)), EudraCT N˚: 2015-001672-22; Depressive study 1: Central IRB (USA_WCG) and local IRB (Belgium (Aalst, Brussel, Duffel), Germany (Berlin, Hamburg, Schwerin), Netherlands (Leiden)), EudraCT N˚: 2014-005182-75; Depressive study 2: Local IRB (Moldova (Chisinau), Poland (Gdansk, Torun, Bialystok, Warszawa), Finland (Helsinki), Latvia (Riga)), EudraCT N˚: 2015-000306-18)) in accordance with the Declaration of Helsinki and the guidelines on Good Clinical Practice. Written consent was obtained from all participants according to local requirements.

**Participants.** Patients were diagnosed with OSA based on the apnea-hypopnea Index (AHI $\geq$ 5 [13]). Insomnia was diagnosed with the Insomnia Severity Index (ISI > 15). The MDD patients met DSM-4 or -5 criteria (using MINI 6.0 or 7.0) and had a score $\geq$30 on the Inventory of Depressive Symptomatology (IDS-C30) or on the Montgomery-Åsberg Depression Rating Scale (MADRS) and a score $\geq$ 4 (markedly ill or worse) on the Clinical Global Impressions Severity Scale (CGI-S).

454 recording nights from 244 subjects were included in the database: 79 nights from 26 healthy participants (Age (Mean±SD): 25±5.8y; F/M ratio: 13/13), 30 nights from 28 patients with OSA (Age (Mean±SD): 54±13y; F/M ratio: 12/16, AHI (Mean±SD): 22±18), 135 nights from 66 patients with insomnia (Age (Mean±SD): 44±14y; F/M ratio: 44/22) and 210 nights from 124 patients with MDD (Age (Mean±SD): 46±13y; F/M ratio: 83/41). To take into consideration the multiple nights from the same subject, the mean sleep parameters of each subject were calculated and only one data point per subject was used for the analysis. Because this paper comprises recording nights from different study protocols, time in bed (TIB) duration varied between studies. To allow comparison between the different studies, all recording nights with a TIB above 388.5 min, corresponding to the shortest TIB of the sample, were cut 388.5 min after Lights-off to fit this minimum duration.

## Study design

All the recording nights included standard PSG with ECG and actimetry recordings.

**Polysomnography.** Multiple PSG recording systems were used in the various studies (Compumedics ProFusion PSG 3; Compumedics Siesta 802a [Compumedics, Abbotsford, Australia]) but all had at least 6 EEG derivations (C3-A2, C4-A1, F3-A2, F4-A1, O1-A2, O2-A1), 2 electro-oculogram electrodes, 2 chin electro-myogram and 2 ECG electrodes. All PSG recorded data were converted into European Data Format (EDF) in order to be processed on a computer screen for visual analysis and scoring [14].

Sleep staging was performed according to the American Academy of sleep medicine rules and the resulting reference classes were obtained by combining N1 and N2 into a single "N1+N2" class while the remaining classes (wake, N3, and REM) were unchanged. The nights from the healthy and the OSA subjects were scored by one experienced scorer per study. The insomnia and the depression studies were scored by an independent expert scorer of the Siesta Group (Vienna, Austria) using the computer-assisted Somnolyzer software [15].

**Cardiac activity from ECG.** Successive inter-beat intervals (R–R intervals) were obtained from the ECG leads from the PSG.

Heart rate data were calculated from R–R intervals as heart rate = 60 / RR (in sec).

**Wrist movement from actimetry.** Non-dominant wrist movement activity was recorded using ActiGraph (Actigraph LLC, Pensacola, FL, USA) activity monitor. Raw data were filtered and accumulated every second. The wrist actimetry was measured through the vector

magnitude of accelerations obtained every second in the 3 dimensions of the space and its value is given in counts per second.

**Somno-Art Software.**   The Somno-Art Software 2.6.0 [3.1.0] analysis was performed on precisely synchronized wrist movement (from actimetry) and cardiac activity (from ECG). Using heart rate at a beat-to-beat resolution and actimetry data at a 1Hz resolution, sleep stage classification (wake, N1+N2, N3, REM) was performed at a 1-s epoch resolution. The latter 1-s epoch classification was merged into 30-s epochs in order to be compared to visual scoring. To do so, the dominant stage (or the first occurring stage, if they were equally represented) was selected. The sleep classification algorithm is based on expert rules associated to Support Vector Machine detectors. Precise data processing methodology is described in Muzet et al. [10].

## Statistical analysis

Overall differences in sleep parameters between the healthy versus the pathology groups and between PSG and Somno-Art Software were evaluated. Mixed model analyses of variance (PROC MIXED) were carried out for each sleep parameter separately (TST, sleep efficiency (SE), WASO, sleep onset latency (SL), REM sleep latency (REML), N1+N2, N3, NREM and REM sleep) and included the fixed factors "group" (healthy and pathology [OSA, insomniac, MDD]), "technology" (PSG, Somno-Art Software), and pathology by technology interaction, and random effect for subject nested within technology. An unstructured covariance matrix was used to model the covariance of within-subject scores. P-values were based on Kenward-Roger's corrected degrees of freedom (Kenward & Roger, 1997).

To further examine the differences observed between the healthy subject and each pathology within each technology, the Mann-Whitney U-tests was performed for each sleep parameter.

Statistical analyses were performed with the SAS statistical software package (SAS Institute Inc., Cary, NC). Significance was set at $p < 0.05$.

## Results

Mixed model analyses were carried out for each sleep parameter investigated. As shown in Table 1, significant differences were observed between the healthy and the pathology group for all sleep parameters except N1+N2 sleep. No significant differences in the overall characterization of sleep parameters were observed between PSG and Somno-Art Software, except for REM sleep. The interaction of "group" by"technology" was non-significant for all sleep parameters.

The differences in sleep parameter between the healthy subjects and each pathology were investigated with Mann-Whitney U-tests within each technology. Fig 1 illustrates sleep parameters measured with PSG (A) and Somno-Art Software (B) for OSA, insomniac and MDD patients compared to healthy subjects.

**Table 1. P-values obtained from the mixed-model with the factors "Group" (Healthy vs pathology), "Technology" (PSG vs Somno-Art Software) and the interaction "Group" *"Technology".**

| Factor | TST | SE | WASO | SL | REML | N1+N2 | N3 | NREM | REM |
|--------|-----|-----|------|-----|------|-------|-----|------|-----|
| Group | <0.0001 | <0.0001 | <0.001 | <0.0001 | <0.01 | 0.16 | <0.0001 | <0.0001 | <0.0001 |
| Technology | 0.98 | 0.98 | 0.8 | 0.78 | 0.91 | 0.56 | 0.69 | 0.4 | 0.047 |
| Interaction | 0.36 | 0.36 | 0.21 | 0.83 | 0.89 | 0.36 | 0.36 | 0.13 | 0.31 |

TST: Total sleep time, SE: Sleep efficiency, WASO: Wake after sleep onset, SL: Sleep onset latency, REML: REM sleep latency.

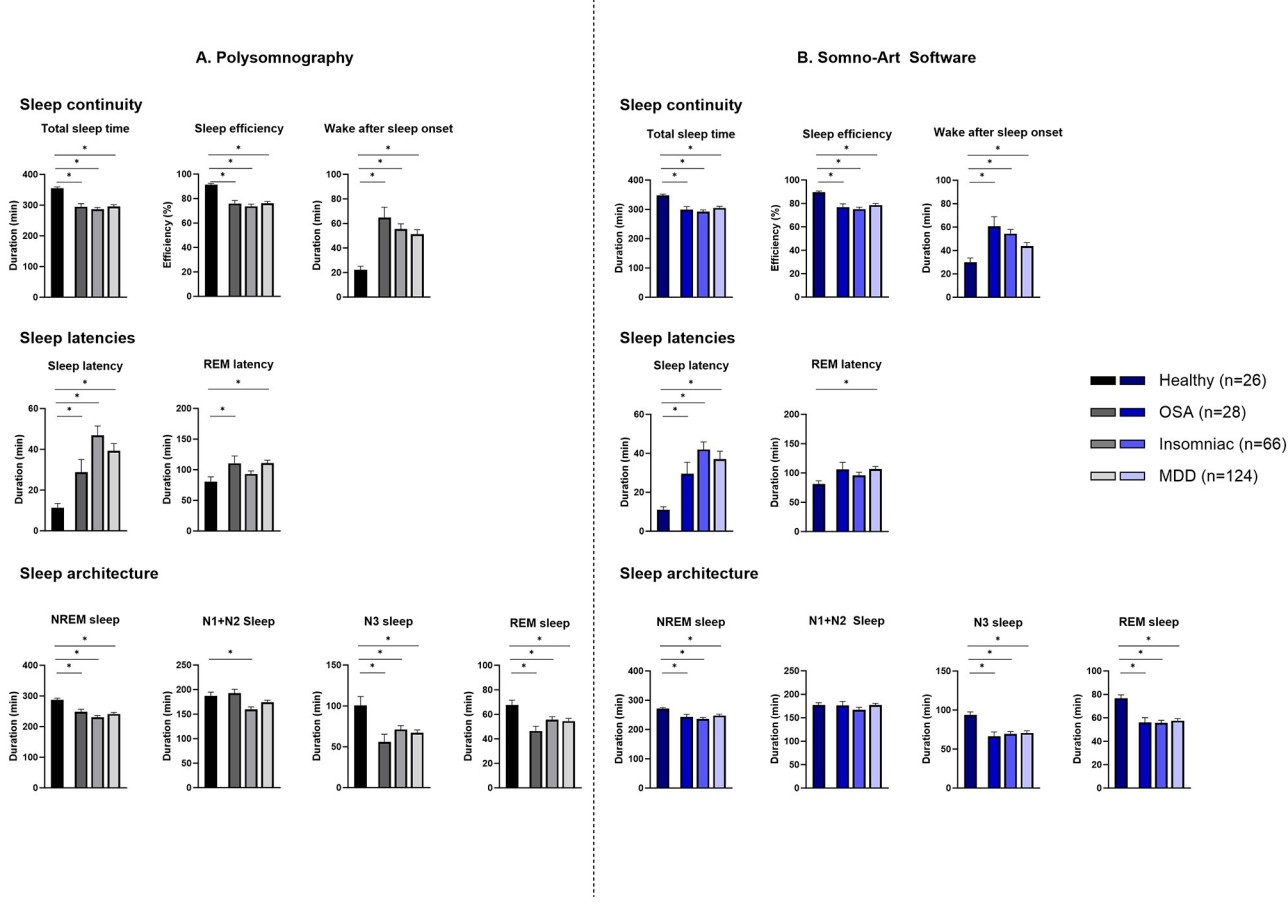

**Fig 1. Mann-Whitney U-test for sleep parameters between healthy subjects and OSA, insomniac or MDD patients obtained from the standard polysomnography (A) and the Somno-Art Software (B).** *p<0.05; (Mean ± standard error).

## Sleep continuity

Both technologies showed lower TST and SE for OSA (difference of the means: PSG: -60 min and -15%; Somno-Art Software: -49 min and -13%), insomniac (PSG: -69 min and -18%; Somno-Art Software: -55 min and -14%) and MDD patients (PSG: -59 min and -15%; Somno-Art Software: -42 min and -11%) compared to healthy subjects. The significant longer WASO observed with PSG for OSA (+43 min), insomniac (+33 min) and MDD (+29 min) patients was also detected with Somno-Art Software (respectively, +31 min, +25 min and +14 min).

## Sleep latencies

Both technologies characterized a longer SL for OSA (PSG: +17 min; Somno-Art Software: +18 min), insomniac (PSG: +35 min; Somno-Art Software: +31 min) and MDD (PSG: +28 min; Somno-Art Software: +26 min) patients compared to healthy subjects. REM sleep latency showed similar pattern between healthy and pathologies with Somno-Art Software and PSG. REM sleep latency was significantly longer in OSA and MDD patients with PSG (+30 min for both pathology). With Somno-Art Software, the difference between healthy and OSA or MDD was significant only for MDD patients (+25 min and +26 min respectively). Both technologies showed no significant differences in REM sleep latency duration between insomniac patients and healthy subjects (PSG: +12 min; Somno-Art Software: +15 min).

## Sleep architecture

Similarly to PSG, Somno-Art Software detected a significant difference in NREM sleep duration between healthy subjects and OSA (PSG: -39 min; Somno-Art Software: -28 min), insomniac (PSG: -57 min; Somno-Art Software: -34 min) and MDD patients (PSG: -46 min; Somno-Art Software: -23 min).

With PSG, N1+N2 sleep was not significantly different between healthy subjects and OSA (+5 min), or MDD (-13 min) patients while insomniac patients showed a significant lower N1+N2 sleep duration (-27 min) compared to healthy subjects. With Somno-Art Software, OSA (-1 min), insomniac (-10 min) and MDD (0 min) patients had no significant different N1+N2 sleep duration compared to healthy subjects.

Both technologies characterized lower N3 sleep in the three pathologies compared to the healthy subjects (OSA: PSG: -44 min and Somno-Art Software: -27 min; insomniac: PSG: -29 min and Somno-Art Software: -24 min; MDD: PSG: -33 min and Somno-Art Software: -23 min).

The strong reduction in REM sleep duration observed between the healthy subjects and the OSA, insomniac and MDD patients was characterized similarly with both technologies (OSA: PSG: -21 min and Somno-Art Software: -21 min; Insomniac: PSG: -12 min and Somno-Art Software: -21 min; MDD: PSG: -13 min and Somno-Art Software: -19 min).

## Discussion

After two validation studies of the Somno-Art Software on healthy subjects [10] and various pathologies [9], the present research is evaluating the performances of the algorithm in identifying differences in sleep parameters observed in various pathologies compared to healthy subjects. Indeed, the Somno-Art Software is intended to be used as an aid for diagnosis of sleep disorders and this paper aims at determining if similar clinical conclusions, such as longer sleep latency in insomniac patients, can equally be drawn when using gold standard PSG or the Somno-Art Software.

A first indication of a correct identification of pathology-induced sleep modification was given by the mixed model analysis. This analysis showed a significant effect of the group factor on each sleep parameter investigated, except for N1+N2 sleep. The type of technology used (PSG, Somno-Art Software) did not affect the sleep parameters. An exception was observed for REM sleep with a 3.6 ± 2.2 min overestimation by Somno-Art Software compared to PSG. However, the differences in REM sleep observed with PSG between the healthy and pathological sleep profiles (e.g. Mann Whitney U-test) was detected by Somno-Art Software. No significant interaction between the type of group (healthy or pathology) and the technology was detected, suggesting that the observed differences in pathologies are independent of the technology used.

The differences within each technology revealed that PSG detected significantly lower TST and SE and significantly longer WASO in OSA patients as compared to healthy subjects. The same differences in sleep continuity parameters were discriminated with the Somno-Art Software. These differences are consistent with the literature that reports fragmented sleep due to frequent sleep apnea leading to lower SE and longer wake duration [16]. Comparable differences in sleep latencies between OSA and healthy subjects were detected with PSG and Somno-Art Software: OSA patients had longer sleep onset latencies and REM sleep latencies than healthy subjects but for REM sleep latency, the difference was significant only with PSG. Somno-Art Software detected similar sleep architecture differences between OSA and healthy subjects than PSG. In accordance with the literature, OSA patients showed lower N3 and REM sleep duration than healthy subjects [17].

In insomniac patients, all 3 sleep continuity parameters investigated, TST, SE and WASO, as well as SL showed comparable differences to healthy subjects when using PSG or Somno-Art Software. Indeed, in line with the literature, insomniac patients showed lower TST and SE and longer WASO and SL than healthy subjects [1]. PSG and Somno-Art Software detected no significant difference in REM sleep latency in insomniac patients compared to healthy subjects. Both PSG and Somno-Art Software detected less NREM, N3 and REM sleep in insomniac patients compared to healthy subjects. PSG detected significantly lower N1+N2 sleep in insomniac patients which was not the case of Somno-Art Software. However, to diagnose insomnia, clinicians focus on the duration of TST, WASO and SL, which were all correctly discriminated with Somno-Art Software.

Both PSG and Somno-Art Software detected lower TST and SE and longer WASO and SL in MDD patients compared to healthy subjects. Both PSG and Somno-Art Software detected significant longer SL and REM sleep latency in MDD patients compared to healthy subjects. Sleep architecture parameters N3 sleep, NREM and REM sleep were significantly lower in MDD patients compared to healthy subjects with PSG and Somno-Art Software. In the literature, MDD patients often present changes in REM pressure with shorter REM sleep latency and more REM sleep which are the key sleep parameters investigated in this pathology [5,6]. This discrepancy between the differences in REM sleep observed in this dataset and the literature may come from the type of dataset used. Indeed, the overall dataset is not stratified by age or gender, demographic factors that are known to impact sleep structure and thus the differences investigated in this paper. This heterogeneity may explain this inconsistency to the literature. Moreover, to allow comparison between study groups, the recording nights were cut at the shortest TIB (388.5 min), removing the last hours of sleep which are characterized by the predominance of REM sleep through the night. However, as the goal of this paper was to evaluate the performance of Somno-Art Software in detecting sleep parameters differences observed with PSG, and not to characterize the sleep profile of the pathologies *per se*, this point should not impact the results presented.

Finally, it is worth noting that this study describes an achievement in the development of wearable sleep stage analysis based on actimetry and cardiac activity. Indeed, this research confirms the ability of Somno-Art Software, a cardiac-based sleep scoring algorithm, to discriminate differences in sleep parameters in patients with disrupted sympatho-vagal balance such as OSA, insomniac or MDD patients [18–20].

## Conclusion

In the present study, both Somno-Art Software and the gold standard PSG showed significant differences in sleep parameters between OSA, insomniac or MDD patients and healthy subjects. Based on the results obtained in this research, the use of the standard PSG or Somno-Art Software leads the clinician to the same diagnostic conclusions about a patient's sleep. Somno-Art opens a new way to measure sleep at home, consistent with PSG, in a less invasive and more time-saving way and easy to use technology which allow for replication.

## Supporting information

**S1 Appendix. Study protocol description.**
(DOCX)

**S1 Dataset.**
(PDF)

## Acknowledgments

We would like to thank the PAREXEL team in Berlin, the sleep disorder centers in Strasbourg and Rouffach for running the experimental studies. We would like to thank Alain Muzet, Valentin Dehouck, Bruno Muller, Julien Barascud and Régis Lengellé for their help in developing the Somno-Art Software algorithm.

## Author Contributions

**Conceptualization:** Laurie Thiesse, Luc Staner, Antoine U. Viola.

**Data curation:** Laurie Thiesse, Henri Comtet, Debora Kirscher.

**Formal analysis:** Laurie Thiesse, Gil Fuchs, Debora Kirscher.

**Funding acquisition:** Jean Yves Schaffhauser.

**Investigation:** Laurie Thiesse, Debora Kirscher, Antoine U. Viola.

**Methodology:** Laurie Thiesse, Luc Staner, Patrice Bourgin, Gil Fuchs, Debora Kirscher, Thomas Roth, Jay B. Saoud, Antoine U. Viola.

**Project administration:** Jean Yves Schaffhauser, Antoine U. Viola.

**Resources:** Patrice Bourgin, Henri Comtet.

**Software:** Gil Fuchs, Debora Kirscher.

**Supervision:** Thomas Roth, Jay B. Saoud, Antoine U. Viola.

**Validation:** Luc Staner, Patrice Bourgin, Gil Fuchs, Jean Yves Schaffhauser, Jay B. Saoud, Antoine U. Viola.

**Writing – original draft:** Laurie Thiesse.

**Writing – review & editing:** Laurie Thiesse, Luc Staner, Patrice Bourgin, Henri Comtet, Gil Fuchs, Debora Kirscher, Thomas Roth, Jean Yves Schaffhauser, Jay B. Saoud, Antoine U. Viola.

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
