## [Decision Letter · Decision Letter 0]

14 Apr 2023

PONE-D-22-35575Somno-Art Software identifies pathology-induced changes in sleep parameters similarly to polysomnography SomnoArt Software for pathological sleep profilesPLOS ONE

Dear Dr. Thiesse,

Thank you for submitting your manuscript to PLOS ONE. After careful consideration, we feel that it has merit but does not fully meet PLOS ONE’s publication criteria as it currently stands. Therefore, we invite you to submit a revised version of the manuscript that addresses the points raised during the review process.One of the reviewer has questioned the statistical analyses used. Please make changes accordingly or explain why the current analysis is statistically appropriate.

We look forward to receiving your revised manuscript.

Kind regards,

Ram A Sharma, MD

Academic Editor

PLOS ONE

Journal Requirements:

2. In the competing interests statement within the manuscript and in the online submission form, please clarify the nature of your competing interests related to your affiliation with PPRS SAS.

"This study was supported by PPRS SAS."

6. We note that one or more of the authors are employed by a commercial company: PPRS SAS

Within your Competing Interests Statement, please confirm that this commercial affiliation does not alter your adherence to all PLOS ONE policies on sharing data and materials by including the following statement: ""This does not alter our adherence to  PLOS ONE policies on sharing data and materials.” (as detailed online in our guide for authors http://journals.plos.org/plosone/s/competing-interests) . If this adherence statement is not accurate and  there are restrictions on sharing of data and/or materials, please state these. Please note that we cannot proceed with consideration of your article until this information has been declared.

7. In your Data Availability statement, you have not specified where the minimal data set underlying the results described in your manuscript can be found. PLOS defines a study's minimal data set as the underlying data used to reach the conclusions drawn in the manuscript and any additional data required to replicate the reported study findings in their entirety. All PLOS journals require that the minimal data set be made fully available. For more information about our data policy, please see http://journals.plos.org/plosone/s/data-availability.

8. We note that you have stated that you will provide repository information for your data at acceptance. Should your manuscript be accepted for publication, we will hold it until you provide the relevant accession numbers or DOIs necessary to access your data. If you wish to make changes to your Data Availability statement, please describe these changes in your cover letter and we will update your Data Availability statement to reflect the information you provide.

Reviewers' comments:

Reviewer's Responses to Questions

**Comments to the Author**

1. Is the manuscript technically sound, and do the data support the conclusions?

Reviewer #1: Yes

Reviewer #2: Yes

2. Has the statistical analysis been performed appropriately and rigorously? 

Reviewer #1: No

Reviewer #2: Yes

3. Have the authors made all data underlying the findings in their manuscript fully available?

Reviewer #1: Yes

Reviewer #2: Yes

4. Is the manuscript presented in an intelligible fashion and written in standard English?

Reviewer #1: Yes

Reviewer #2: Yes

5. Review Comments to the Author

Reviewer #1: The main problem of the manuscript is a poor statistical analysis. From my point of view, the authors should analyse sleep parameters using a mixed model ANOVA taking into account the technique of sleep evaluation (PSG vs. SomnoArt) and the clinical status (Healthy subjects vs. OSA, Insomnia, and MDD patients respectively) rather than multiple comparisons using Mann-Whitney U-test. The absence of significant 'technique' effect and significant 'technique x clinical status' interactions would be more demonstrative results.

The results section and the discussion must therefore be rewritten and a synthesis of the results should also be presented in one or two tables.

Reviewer #2: This is an interesting paper which highlighted the possibilities of using Somno-Art in the relevant setting. A concise and useful paper indicating that there may indeed be a place for Somno-Art in such a setting.

6. PLOS authors have the option to publish the peer review history of their article (what does this mean?). If published, this will include your full peer review and any attached files.

Reviewer #1: **Yes: **Dr Olivier COSTE (Lyon), MD-PhD

Reviewer #2: No

---

## [Author Response · Author response to Decision Letter 0]

31 May 2023

5. Review Comments to the Author

Reviewer #1: The main problem of the manuscript is a poor statistical analysis. From my point of view, the authors should analyse sleep parameters using a mixed model ANOVA taking into account the technique of sleep evaluation (PSG vs. SomnoArt) and the clinical status (Healthy subjects vs. OSA, Insomnia, and MDD patients respectively) rather than multiple comparisons using Mann-Whitney U-test. The absence of significant 'technique' effect and significant 'technique x clinical status' interactions would be more demonstrative results.

The results section and the discussion must therefore be rewritten and a synthesis of the results should also be presented in one or two tables.

Answer: Thank you for your review. We understand your concern with the statistical analysis. We decided not to include the mixed model analysis, as the goal of this publication was to check if Somno-Art can detect changes in sleep parameters observed in sleep disorders compared to healthy. Comparing the performances of Somno-Art to PSG in healthy subjects and pathologies, has already been published in previous publication.

However, as you suggested, we integrated the mixed model analysis to give an overview. The fact that the interactions ‘group x technology’ were non-significant suggests that the differences in pathologies are independent of the technology used. In a second step we kept the Mann-Whitney U-tests to have an in-depth analysis of the differences we are interested in (healthy vs OSA, healthy vs INS, healthy vs MDD).

Reviewer #2: This is an interesting paper which highlighted the possibilities of using Somno-Art in the relevant setting. A concise and useful paper indicating that there may indeed be a place for Somno-Art in such a setting.

Answer: We thank reviewer 2 for the review.

---

## [Decision Letter · Decision Letter 1]

4 Sep 2023

Somno-Art Software identifies pathology-induced changes in sleep parameters similarly to polysomnography

PONE-D-22-35575R1

Dear Dr. Thiesse,

We’re pleased to inform you that your manuscript has been judged scientifically suitable for publication and will be formally accepted for publication once it meets all outstanding technical requirements.

Kind regards,

Ram A Sharma, MD

Academic Editor

PLOS ONE

Additional Editor Comments (optional):

Reviewers' comments:

Reviewer's Responses to Questions

**Comments to the Author**

1. If the authors have adequately addressed your comments raised in a previous round of review and you feel that this manuscript is now acceptable for publication, you may indicate that here to bypass the “Comments to the Author” section, enter your conflict of interest statement in the “Confidential to Editor” section, and submit your "Accept" recommendation.

Reviewer #1: All comments have been addressed

Reviewer #2: All comments have been addressed

2. Is the manuscript technically sound, and do the data support the conclusions?

Reviewer #1: Yes

Reviewer #2: (No Response)

3. Has the statistical analysis been performed appropriately and rigorously? 

Reviewer #1: Yes

Reviewer #2: (No Response)

4. Have the authors made all data underlying the findings in their manuscript fully available?

Reviewer #1: Yes

Reviewer #2: (No Response)

5. Is the manuscript presented in an intelligible fashion and written in standard English?

Reviewer #1: Yes

Reviewer #2: (No Response)

6. Review Comments to the Author

Reviewer #1: The authors have taken into account my remarks dealing with statistical analysis. They have also widely improved the general presentation of their paper.

Therefore, it is now ready for publication.

Reviewer #2: (No Response)

7. PLOS authors have the option to publish the peer review history of their article (what does this mean?). If published, this will include your full peer review and any attached files.

Reviewer #1: **Yes: **Olivier Coste (MD, PhD)

Reviewer #2: No

---

## [Editor Report · Acceptance letter]

11 Oct 2023

PONE-D-22-35575R1 

Somno-Art Software identifies pathology-induced changes in sleep parameters similarly to polysomnography 

Dear Dr. Thiesse:

I'm pleased to inform you that your manuscript has been deemed suitable for publication in PLOS ONE. Congratulations! Your manuscript is now with our production department. 

Kind regards, 

on behalf of

Dr. Ram A Sharma 

Academic Editor

PLOS ONE